# Low-Temperature Multiple Micro-Dispensing on Microneedles for Accurate Transcutaneous Smallpox Vaccination

**DOI:** 10.3390/vaccines10040561

**Published:** 2022-04-04

**Authors:** Sang-Gu Yim, Yun-Ho Hwang, Seonyeong An, Keum-Yong Seong, Seo-Yeon Kim, Semin Kim, Hyeseon Lee, Kang-Oh Lee, Mi-Young Kim, Dokeun Kim, You-Jin Kim, Seung-Yun Yang

**Affiliations:** 1Department of Biomaterials Science (BK21 Four Program), Life and Industry Convergence Institute, Pusan National University, Miryang 50463, Korea; sg.yim0425@gmail.com (S.-G.Y.); seonyeong510@gmail.com (S.A.); ky.seong0124@gmail.com (K.-Y.S.); hyeslee96@gmail.com (H.L.); 2Division of Infectious Disease Vaccine Research, Center for Vaccine Research, National Institute of Infectious Diseases, National Institute of Health, Korea Disease Control and Prevention Agency, Cheongju 28159, Korea; yunho1129@korea.kr (Y.-H.H.); tjdus9284@korea.kr (S.Y.K.); mkim2612@korea.kr (M.-Y.K.); dickykim@korea.kr (D.K.); 3SNVIA Co., Ltd., Hyowon Industry-Cooperation Building, Busan 46241, Korea; smkim@snvia.com (S.K.); koled@snvia.com (K.-O.L.)

**Keywords:** smallpox vaccination, microneedle, live virus, transcutaneous inoculation

## Abstract

Smallpox is an acute contagious disease caused by the variola virus. According to WHO guidelines, the smallpox vaccine is administrated by scarification into the epidermis using a bifurcated needle moistened with a vaccine solution. However, this invasive vaccination method involving multiple skin punctures requires a special technique to inoculate, as well as a cold chain for storage and distribution of vaccine solutions containing a live virus. Here, we report a transcutaneous smallpox vaccination using a live vaccinia-coated microneedle (MN) patch prepared by a low-temperature multiple nanoliter-level dispensing system, enabling accurate transdermal delivery of live vaccines and maintenance of bioactivity. The live vaccinia in hyaluronic acid (HA) solutions was selectively coated on the solid MN tips, and the coating amount of the vaccine was precisely controlled through a programmed multiple dispensing process with high accuracy under low temperature conditions (2–8 °C) for smallpox vaccination. Inoculation of mice (BALB/C mouse) with the MN patch coated with the second-generation smallpox vaccine increased the neutralizing antibody titer and T cell immune response. Interestingly, the live vaccine-coated MN patch maintained viral titers at −20 °C for 4 weeks and elevated temperature (37 °C) for 1 week, highlighting improved storage stability of the live virus formulated into coated MN patches. This coated MN platform using contact dispensing technique provides a simple and effective method for smallpox vaccination.

## 1. Introduction

Smallpox, one of the deadliest diseases in human history, is an acute infectious disease caused by the variola virus [1,2,3,4,5]. After an asymptomatic incubation period of 10 to 14 days (range 7 to 19 days), the common symptoms of smallpox appear, including high fever, head and body aches, and a distinctive rash [6,7,8]. It is a highly contagious disease with high rates of death (approximately 30%) [9,10]. Following the success of a worldwide vaccination program against smallpox, the World Health Organization (WHO) declared smallpox eradication in 1980 and advised all countries to stop routine smallpox vaccination [4,11,12]. Although routine vaccination against smallpox is no longer performed, smallpox vaccines are still produced and stockpiled to guard against bioterrorism and biological warfare [6,13]. 

Currently, smallpox vaccines derived from cell culture are administered by skin scarification on the deltoid muscle or the posterior muscle of the arm using the multiple-puncture technique with a bifurcated needle [14,15,16,17,18]. For smallpox vaccination, 15 punctures are recommended, and a trace of blood should appear at the vaccination site. However, this invasive vaccination method using bifurcated needles has raised safety issues, such as risk of injury, disease transmission, and generation of biohazardous sharp wastes. It also requires a special technique to inoculate, and a cold chain to store and distribute the vaccine solution containing live viral species (vaccinia virus) [19,20].

To improve scarring and pain caused by inoculation, alternative administration methods for the delivery of smallpox vaccines have been developed [17]. The injection method using a syringe needle through the subcutaneous and intramuscular route is a common method of vaccination, but the effectiveness of a smallpox vaccination using the injection technique was lower than that of the scarification method [14,17,21,22]. Alternatively, the jet injection using a high-speed stream of fluid has been used for intradermal delivery of smallpox vaccines [23,24]. This method was beneficial for rapid vaccination, but it presented limitations, such as pain during injection, perforation, and risk of infection [17,25,26].

Recently, transdermal vaccination using microneedle (MN) technology has attracted great attention to enhance the effectiveness of vaccines in a minimally invasive manner [27,28,29,30,31,32,33,34]. MN-mediated transdermal delivery of vaccines including viral, bacteria, and mRNA showed better immune boosting compared to conventional bolus vaccination [35,36]. Due to enormous potential such as painless administration, enhanced thermostability, dose-sparing capacity, and self-administration, MN-based transdermal drug delivery system is recognized as a highly promising platform of vaccination [28,29,30,31,37]. Among various MN types, coated MNs prepared by dip coating, spray coating, or ink-jet coating methods have been widely investigated due to its fast drug delivery kinetics and simple fabrication process [38,39,40,41,42,43]. However, the limited coating amount and difficulty of precise and selective drug coating on MN shafts remain significant challenges for developing MN-based inoculation methods. In addition, the stability of bioactive agents such as live vaccines during MN fabrication or storage is an important factor to be considered for vaccination and stockpiling [44].

Herein, we report a transcutaneous smallpox vaccination using a live vaccinia-coated MN patch prepared by low-temperature multiple nanoliter-level dispensing system. The live vaccinia in hyaluronic acid (HA) solutions was selectively coated on the solid MN array, and the coating amount of the vaccine was precisely controlled through programmed multiple dispensing process with high accuracy for the standard dose (2.5 × 10^5^ plaque forming unit (PFU)) of smallpox vaccination. To minimize loss of viral vaccine activity and develop low-temperature MN vaccine system (conceptually illustrated in Figure 1), fabrication and packaging of the MN vaccine patches were performed in a cold room at low temperatures (2–8 °C). The stability and activity of vaccine coating formulation was evaluated depending on the working temperature. The safety and efficacy of smallpox vaccine-coated MN patches were evaluated using experimental animals following applications to the skin.

## 2. Materials and Methods

### 2.1. Preparation of Coating Solutions

The second-generation vaccinia virus vaccine (CJ-50300) derived from MRC-5 cells using a vaccinia virus strain (ATCC VR-118) [45,46] was produced in HK inno.N Corp. (Seoul, Korea) and was provided by the Korea Disease Control and Prevention Agency (KDCA). To prepare a coating solution, 1 wt% sodium hyaluronate (HA; SNVIA Co., Ltd., Busan, Korea) used as a viscous enhancer was added to the smallpox (vaccina virus) vaccine solution (2.0 ± 0.1 × 10^7^ PFU/mL in phosphate-buffered saline (PBS; Biosesang, Suwon, Korea)) and dissolved at 4 °C for 12 h. The HA was analyzed and fractionated by gel permeation chromatography (GPC) using the following system: Waters Alliance e2695 system, Waters 2414 Refractive index detector, and Ultrahydrogel™ Linear column (7.8 × 300 mm) (USA). The mobile phase was 50 mM NaCl, and the flow rate was 0.7 mL/min. The HA solution (40 μL) with a 1 mg/mL concentration was measured at 30 °C. The viscosity of the HA solution was measured with a viscometer (microVISC™; RheoSense, Inc., San Ramon, CA, USA). The coating solutions were stored at 4 °C and 25 °C, respectively, to examine the effect of working temperature on vaccine stability through plaque assay.

### 2.2. Fabrication of Coated MNs Using Contact Dispensing System

MN arrays (8 × 8 MNs in 1.4 cm × 1.4 cm) produced by injection molding with medical-grade thermoplastic polymers were provided by SNVIA Co., Ltd. (Busan, Korea) and used for the base MN for coating process. The morphology of the MN arrays was examined using an optical microscope (Eclipse TS100; Nikon, Tokyo, Japan), and the dimensions of each MN were measured using image analysis software (Image J). The base MN array was coated by using high-precision contact dispensing system (HSS-01; SNVIA Co., Ltd., Busan, Korea), consisting of machine vision, 3-axis robotic dispensing machine, and control panel. The MN coating was carried out in a modular temperature-controlled room (room temperature (25 °C) to 0 °C). After plasma treatment of the base MN by using a surface treatment system (STM-01; SNVIA Co., Ltd., Busan, Korea) for selective coating [47], MN arrays were placed on the automated dispensing cell. After checking the inclination and height of the base MN through the vision system, automatic coating on MN tips was performed with a single dispensing nozzle. The coating amount on the MN patch was controlled by adjusting dispensing pressure, discharge time, and the number of coatings. To evaluate quantitative coating properties by contact dispensing system, HA solution (1 wt%) with 1 mg/mL rhodamine 6G (absorption wavelength: 440~570 nm; Sigma-Aldrich, St. Louis, MO, USA) used as a model drug was repeatedly coated on the base MN (1, 3, 5 times) [48]. The coating process was monitored with a digital microscope (DINO-Lite; AnMo Electronics Co., Taiwan). After completely dissolving the dried coating layer by immersion in 70% ethanol for 30 min, the amount of the rhodamine 6G coated on MN was quantified through UV-vis absorption spectroscopy with a microplate reader (AMR-100; Allsheng Co., Hangzhou, China). Considering coating volume (2 μL) on the base MN array (64 MNs), 1 wt% HA coating solution containing vaccina virus vaccine (2.0 ± 0.1 × 10^7^ PFU/mL) was coated on the base MN multiple times (1, 3, 5 times) to match the standard dose of smallpox vaccination (2.5 × 10^5^ PFU). The coated MNs were subsequently dried for 1 h at 4 °C and 25 °C, respectively. The vaccine stability was measured through plaque assay following the complete dissolution of dried coating layer by immersion in 400 μL Dulbecco’s modified Eagle’s medium (DMEM) solution of 12-well plate for 30 min.

### 2.3. Plaque Assay

Plaque assays for vaccine stability testing were performed as follows [49]: Vero cells purchased from the American Type Culture Collection (ATCC, CCL-81^TM^) and were seeded in a 12-well plate at a density of 2 × 10^5^ cells/well and incubated for 24 h. The culture media was removed and washed with 1 mL PBS solution twice; each specimen was 10-fold serially diluted in a maintenance medium containing 2% fetal bovine serum (FBS). With each change of pipette tip, 400 µL of the diluted specimens were dispensed into the designated wells and incubated at 37 °C and 5% CO_2_ in a humidified atmosphere for 1 h. Overlay medium was prepared by combining equal amounts (1:1) of DMEM supplemented with 4% FBS and 2% carboxymethyl cellulose (CMC, Sigma-Aldrich, St. Louis, MO, USA) in distilled water. After 1 h, 1.5 mL of overlay medium was added to each well of the titration plate, and it was incubated at 37 °C in a 5% CO_2_ incubator for 3 days. The overlay medium was carefully aspirated, and cells in each well were gently washed with PBS. Crystal violet mixture solution was prepared by 0.6% crystal violet solution (Sigma-Aldrich, St. Louis, MO, USA) in a 37% formaldehyde solution (Sigma-Aldrich, St. Louis, MO, USA), ethanol, and distilled water. 500 µL of crystal violet mixture solution was added to each well and incubated at room temperature for 30 min. Virus dilution factors representing 5–100 plaques per well were identified, and the virus titer was calculated by counting the plaques.

### 2.4. Animals and Care

The animal protocol used in this study was reviewed and approved based on ethical procedures and scientific care by the Pusan National University-Institutional Animal Care and Use Committee (PNU-IACUC). Eight-week-old female BALB/C mice (17–22 g) were purchased from Samtako Bio Korea (Korea) and handled at the Pusan National University Laboratory Animal Resources Center (Korea), which is accredited by the Korea Food and Drug Administration according to the National Institutes of Health guidelines. Mice were housed in cages under a 12 h light/dark cycle, with a constant temperature of 23 ± 1 °C. All experimental animal procedures were approved by the Ethics Committee of Pusan National University (approval code: PNU-2020-2548).

### 2.5. In Vivo Vaccination Test

One day before vaccination, hairs on the dorsal skin (2 × 2 cm^2^) of mice were removed using a depilatory cream (Nair™; Church and Dwight, New York, NY, USA) after hair removal with clippers. In vivo tests were conducted in a total of 3 groups (10 mice per group): control (no treatment), only HA-coated MN (MN vehicle), and vaccine-coated MN (MN vaccine). The MN patches with an adhesive backing layer (NEODerm; Everaid, Yangsan, Korea) were applied on the shaved dorsal skin of the mice for 2 min and then affixed with a bandage (High fix, SM Medical Co., Daejeon, Korea) for 24 h. The morphology of the vaccine-coated MN before and after insertion was observed by scanning electron microscope (SEM; Hitachi S-4700, Tokyo, Japan). After collecting MN patches (*n* = 7) used for vaccine administration, the titers of the virus were analyzed by plaque assay after dissolution procedure described above. Serum samples were collected from the submandibular vein at 3 and 6 weeks (*n* = 7), and spleen samples were isolated from the mouse at 12 weeks (*n* = 10) after inoculation.

### 2.6. Analysis of Neutralizing Antibody Responses after Vaccination

Plaque reduction neutralization test (PRNT) was performed to analyze neutralizing antibodies to the smallpox vaccine [49]. PRNT titers were determined using the highest serum dilution that inhibited more than 50% of plaques than the number of plaques in the absence of test serum. Serum samples were heat-inactivated for 30 min at 56 °C prior to use. Inactivated serum samples and 2× serial dilutions of virus samples containing 100 PFU were mixed and incubated for 1 h at 37 °C. Vero cells in 12-well plates (approximately 5 × 10^5^ cells/well) were treated with the cultured virus-serum mixture. The final overlay was performed using 0.2% low melting point agarose in Opti-MEM™ medium containing 2% FBS. After incubating the cells at 37 °C for 3 days, the cells were fixed and stained with a crystal violet mixture, and then plaques were counted.

### 2.7. Enzyme-Linked Immunosorbent Spot (ELISPOT) Assay

ELISPOT was measured using splenocytes isolated from mice at 12 weeks after vaccination. 96-well polyvinylidene difluoride (PVDF)-backed microplate coated with a monoclonal antibody specific for mouse interferon gamma (IFN-γ) was blocked by complete RPMI for 2 h. Splenocytes (5 × 10^5^ cells/well) from which red blood cells (RBCs) were removed by RBC lysis solution were stimulated with CJ-50300 0.1 multiplicity of infection (MOI) for 18 h at 37 °C incubator. The plate from which cells were removed was washed with wash buffer, and biotinylated monoclonal antibody specific for mouse IFN-γ was treated in each well and incubated for 2 h at room temperature on a rocking platform. After washing, the plates were incubated with Streptavidin conjugated to Alkaline Phosphatase for 2 h. Incubated plates were developed using 5-bromo-4-chloro-3-indolyl phosphate/Nitro blue tetrazolium (BCIP/NBT) Substrate for 1 h at room temperature. Spots were counted using an Immunospot reader (Cellular Technology Limited., Shaker Heights, OH, USA).

### 2.8. Long-Term Storage Tests

For this study, we additionally supplemented trehalose in the coating solution after checking its concentration-dependent toxicity to Vero cells. For cytotoxicity tests, Vero cells were seeded in a 24-well plate at a density of 2 × 10^4^ cells/well and incubated for 24 h. The seeded cells were treated with different trehalose concentrations (0, 1, 5, 10 and 15% (*w*/*v*); Sigma-Aldrich, St. Louis, MO, USA) for 24 h. The cells were supplemented with 10 μL of WST-1 reagent (EZ-Cytox; DogenBio, Seoul, Korea) and incubated for 1–2 h. The cell viability was measured based on the colorimetric OD value at 450 nm using a microplate reader. 

To investigate the storage stability depending on storage temperature, vaccine-coated MNs (target dose: 2.5 × 10^5^ PFU) were prepared with a coating solution (1 wt% HA) supplemented with trehalose (0, 1, 5, 10 and 15%(*w*/*v*)), dried for 30 min, vacuum-sealed at −4 °C, and then stored at −20 °C and 37 °C, respectively, for 4 weeks. The titers of MN samples were measured through plaque assay at predetermined time points (1, 7, 14, 21, 28 days).

### 2.9. Statistical Analysis 

Data were analyzed using Student’s t-test with Excel software (Microsoft Corp., Redmond, WA, USA) and expressed as the mean ± standard deviation (SD) of independent experiments. *p*-values < 0.05 indicate statistically significant differences.

## 3. Results and Discussion

### 3.1. Preparation of Vaccine Coating Solutions

To prepare a vaccine-coated MN by contact dispensing, excipient as a viscous enhancer is needed to achieve uniform and precise coating on MN shafts. HA was selected as a water-soluble excipient to modulate the viscosity of vaccine coating solutions. HA, a non-toxic and biodegradable linear polysaccharide, has been widely used as a drug delivery carrier and approved by the Food and Drug Administration (FDA) as inactive material [50]. We used a medical-grade HA (M_n_: 430 kDa), confirmed by GPC measurement (Figure 2a). According to optimal viscosity range (20~5000 cP) of a coating solution for contact dispensing recommended by the manufacturer, HA was prepared as a 1 wt% aqueous solution with a viscosity of ~500 cP (Figure 2b).

After preparing a coating solution by thawing a lyophilized virus vaccine, adding HA, and mixing for 12 h, we investigated the influence of storage temperature on vaccine stability by measuring viral titers through plaque assay. As shown in Figure 2c, the vaccine activity was not significantly affected by the addition of HA or storage temperature at the early time point (12 h). However, in the absence of HA, prolonged exposure to ambient temperature (25 °C) reduced the titer of the virus solution by approximately 60%. Interestingly, the addition of HA to vaccine solutions improved the vaccine stability at 25 °C, possibly due to the formation of a HA network around the virus [51]. When stored at 4 °C, the coating solution exhibited better stability (~85% compared to virus solution after thawing), highlighting the importance of working temperature during the fabrication of live vaccine-coated MN. We additionally investigated the molecular weight (MW) effect of HA in vaccine stability by measuring the virus titers following addition of low MW HA (100 kDa) to the vaccine solution and mixed for 12 h at 4 °C. There is no significant difference in virus titers between the two groups (Appendix A).

### 3.2. Selective and Precise Vaccine Coating on MN Tips Using Low-Temperature Multiple Dispensing System

For accurate dose control, selective and uniform coating of drug solution on the MN shaft without base contamination is desirable [39]. To achieve selective and precise coating of vaccine solution on base MN arrays, we used a contact dispensing method after plasma treatment of base MN arrays. The base MN arrays (8 × 8 MNs/patch) was prepared by injection molding, and each bullet-shaped MN had a base diameter of 350 μm and a height of 900 μm. As shown in Figure 3a, the contact dispensing technique accurately transferred single microdroplets of 1 wt% HA solution with rhodamine dye from the nozzle to the MN tip and provided multiple coating after drying previous layers. Thus, the target amount of drug to be delivered into skin can be easily adjusted according to the number of dispensing coatings. To confirm controllable coating in drug amount, we performed multiple coating of HA solutions containing red dyes on the MN tips. The coating solution was uniformly coated on MN shaft only (Figure 3b) and the coating mass increased linearly with the number of coatings (Figure 3c), highlighting the precise dose control of the coated MN fabricated by the programmed multiple dispensing process. 

We next investigated the effect of working temperature on vaccine stability during the manufacturing process (coating and drying) of vaccine coated MNs. To offer constant temperature conditions during fabrication process, MN coating and drying were performed in a modular cold room, as shown in Figure 3d. After preparing HA coating solution containing vaccina virus vaccine (titer: 2.0 ± 0.1 × 10^7^ PFU/mL) at 4 °C, vaccine solution was coated on the base MN array 5 times (approximately 10 μL) at different coating (4 °C and 25 °C) and drying (4 °C and 25 °C) temperatures (Figure 3e). The virus titer measured by plaque assay following complete dissolution of coated layer on the MN array decreased after exposure at normal room temperature (25 °C). Especially, exposure to ambient temperature (25 °C) during coating process showed a greater adverse effect on virus activity. This result suggests that minimizing the temperature change during the manufacturing of coated MNs is beneficial in maintaining the titer of the vaccine virus. However, the virus titer of vaccine-coated MN decreased by about 45% even under low-temperature conditions due to stress induced during drying. 

To confirm the accurate dose control with the vaccine coating solution, the virus titers of the coated MNs were measured after programmed multiple coatings (1, 5, and 10 times) (Figure 3f). As anticipated, mean virus titers increased corresponding to the number of coatings, thus enabling precise control of target dose. 

### 3.3. In Vivo Smallpox Vaccination Tests Using Live Vaccina-Coated MN Patches

To evaluate the efficacy of MN-mediated transcutaneous smallpox vaccination, BALB/C mice were immunized with live vaccina vaccine-coated MN patches. Considering the standard dose (2.0~2.5 × 10^5^ PFU) for smallpox vaccination [52], the same virus titer of vaccine was coated on base MN arrays by 5 times contact dispensing (total dispensing volume: ~10 μL) using vaccine coating solutions (titer: 2.0 ± 0.1 × 10^7^ PFU/mL). HA-coated MN arrays were used as a control (MN vehicle). After inoculation, serum samples were collected at 3 and 6 weeks to examine neutralizing antibodies against the virus through PRNT (Figure 4a).

The MN patches with adhesive backing layer were applied to the shaved dorsal area of the mouse by thumb pressure for 2 min (Figure 4b). After 24 h, the applied MN patches were removed and then characterized to examine vaccine delivery efficiency through SEM measurement and virus titer analysis. As shown in Figure 4c, the coated layer formed on the top of MN was clearly dissolved after application. To investigate the amount of virus vaccine remaining on the surface of the MN patch, we compared the virus titers of vaccina-coated MN patches (MN vaccine) before and after application. Only 1.4% of virus (1.70 ± 0.48 × 10^3^ PFU) was measured from MN patch after administration compared to pre-inoculation viral titers (1.20 ± 0.05 × 10^5^ PFU), indicating accurate intradermal delivery of the vaccine (Figure 4d). Although the titer of MN vaccine was decreased by 60% compared to the vaccine solution (2.0 ± 0.1 × 10^5^ PFU/10 μL), MN vaccine with the same potency can be prepared by additional coating process or by increasing the viral titers of the coating solution.

To examine the immune response after vaccination using vaccina (second-generation smallpox vaccine)-coated MN patches, we measured neutralizing antibody against the vaccinia virus through PRNT assay. As shown in Figure 4e, neutralizing antibody titer was increased at 3 weeks and 6 weeks in the MN-mediated inoculation group (MN vaccine) compared to the control group (no treatment) or vehicle group (MN vehicle). In addition, we confirmed that the smallpox vaccine inoculated by MN patch induced a T-cell immune response (Figure 4f). After application with vaccine-coated MN patches, cutaneous reactions were observed at the inoculation site in the first week, but almost all wounds were healed with minimal scarring by the second week (Appendix A). However, to obtain an immune response similar to that induced after standard scarification vaccination (Appendix A), it is necessary to improve the immunization efficacy of the MN vaccine by adjusting the coating amount of the virus vaccine. Considering potential side effects of replicating vaccinia vaccines used in this study [53], highly attenuated or peptide-based vaccines [54,55,56,57] would be alternative to develop a safer smallpox vaccination. Although attenuated non-replicating vaccines require more viral titers for effective vaccination, our MN platform enabling multiple coatings of vaccine solutions can be applicable for non-replicating vaccines. In addition, the high-precision coating ability of target molecules allows the coated MN system to be applied for a skin prick test to check allergic reactions to various allergens [58].

### 3.4. Storage Stability Test of MN Vaccines

We next examined the stability of MN vaccines depending on storage temperature. For this study, trehalose, a commonly used as a cryoprotectant and stabilizing agent [46,59,60], was additionally supplemented to the vaccine coating solutions after confirming its allowable concentration range to use as an excipient. When trehalose was treated to Vero cells at different concentrations (0, 1, 5, 10, and 15% *w*/*v*), cytotoxic effect was not significant at a concentration of less than 5% trehalose, resulting in a cell viability of more than 90% compared to the non-treated group (Figure 5a). We further investigated the vaccine stability in coating solutions as a function of trehalose concentration. As shown in Figure 5b, there were no significant differences in virus titers depending on trehalose concentrations. However, since trehalose could reduce the stress caused by drying or temperature changes [61], the highest concentration (5%) of trehalose in the range not showing cytotoxicity was added to the coating solution. 

To investigate the storage stability depending on storage temperature, MN vaccine patches were prepared with a coating solution containing the combination of HA (1%) and trehalose (5%) at 4 °C, stored either at −20 °C and 37 °C for 4 weeks, and measured the viral titers after complete dissolution of the coated layer (Figure 5c). For MN vaccines stored at −20 °C, virus titers decreased slightly at the initial time point of storage, but there was no statistically significant difference for 4 weeks. In addition, the virus titer of MN vaccine was maintained for 1 year when it stored at −20 °C (data not shown). In case of MN vaccines stored at 37 °C, it showed a similar virus titer during storage of 1 week. Thereafter, however, the virus titer gradually decreased, showing 44% reduction in titer at 4 weeks. Although the MN vaccine showed short-term stability at elevated temperature for 1 week, it is advantageous for facilitating mass vaccination campaigns, especially in developing countries [62].

## 4. Conclusions

We demonstrated the feasibility of transcutaneous smallpox vaccination using live vaccinia-coated MN patches prepared via contact dispensing technique. This new coating method for coated MNs enabled accurate transfer of nanoliter-level single microdroplets containing a smallpox vaccine from the dispensing nozzle to the base MN tip and offered multiple deposition of coating-solutions after drying of previous coating layer, thereby achieving precise control of target dose. HA was used as an excipient in the coating solution to increase coating uniformity and virus viability during MN fabrication process. In stability tests, we found that constant temperature condition at low temperature (2–8 °C) is beneficial to prevent the reduction of virus viability during MN manufacturing process (coating and drying). The smallpox vaccine-coated MN patch prepared by programmed multiple dispensing of vaccine solutions showed accurate intradermal delivery (1.4% virus residual in MN patch after inoculation) and induced both humoral and cell-mediated immune responses in experimental animals (BALB/C mice), suggesting the potential for MN-mediated smallpox vaccination. Since the titer of MN vaccine patches (1.20 ± 0.05 × 10^5^ PFU) was less than 50% compared to the standard dose (2.5 × 10^5^ PFU) for smallpox vaccination for human [46], additional dose-dependent efficacy tests for smallpox vaccination should be investigated. In storage stability tests, the vaccine-coated MN patch maintained viral titers at −20 °C for 4 weeks and elevated temperature (37 °C) for 1 week, highlighting improved storage stability of live virus formulated into coated MN patches. Since the coating amount and position to each MN can be easily controlled via programmed coating process, this coated MN platform would be suitable for transcutaneous immunization of a diverse range of vaccines.

## Figures and Tables

**Figure 1 vaccines-10-00561-f001:**
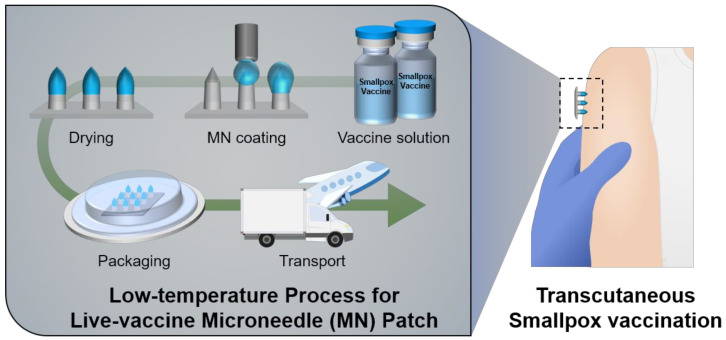
Schematic illustration showing the low-temperature process to apply live vaccinia-coated MN patches for transcutaneous smallpox vaccination.

**Figure 2 vaccines-10-00561-f002:**
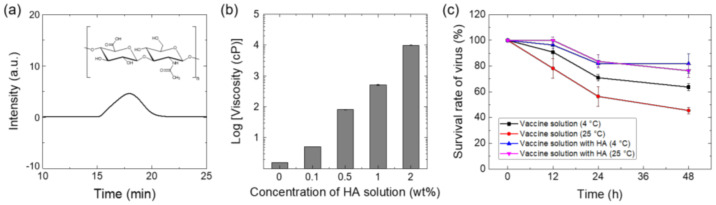
(**a**) Gel permeation chromatography (GPC) of HA used in the coating solution as a viscous enhancer. (**b**) The viscosity of HA solutions with different concentrations. (**c**) Effect of the excipient (viscous enhancer) and storage temperature on the activity of smallpox vaccine (vaccinia virus) in solutions.

**Figure 3 vaccines-10-00561-f003:**
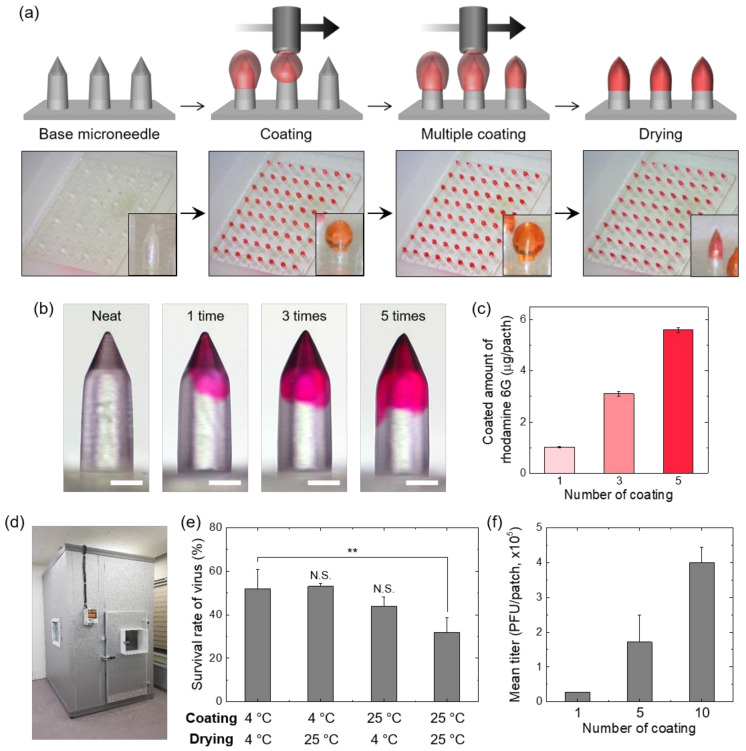
(**a**) Fabrication procedure for coated MN arrays (64 MNs/patch) using multiple dispensing system. Rhodamine red dye was added to the HA coating solution for visualization. (**b**) Optical microscopic images showing the side view of MN after multiple coating (0, 1, 3, and 5 times) of HA solutions with rhodamine dye (scale bar, 200 μm). (**c**) Coated amount of rhodamine dye in the MN array depending on the number of coatings. (**d**) Photo showing a modular cold room capable of maintaining low temperatures (2–8 °C). The high-precision contact dispensing system for MN coating was installed in this cold room. (**e**) Effect of ambient temperature on vaccine stability during the manufacturing process (coating and drying) of vaccine coated MNs (*n* = 6). Relative titer compared to the vaccine coating solution was examined through plaque assay after dissolving the coated layer by immersing the MN array in PBS. (**f**) Titer of vaccine-coated MN patches prepared by different coating times (*n* = 6). Statistical significance compared to the coating at 4 °C and drying at 4 °C sample was determined by a *t*-test (** *p* < 0.01).

**Figure 4 vaccines-10-00561-f004:**
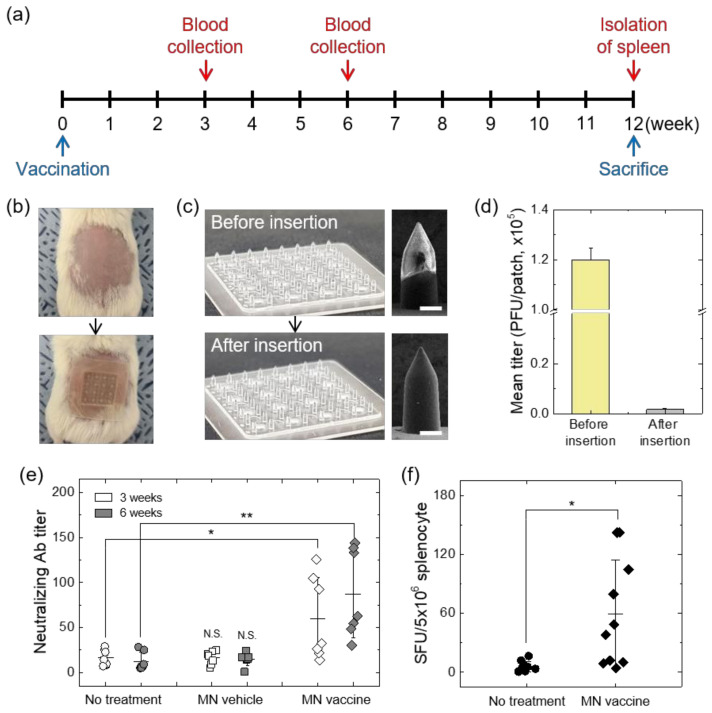
(**a**) Experimental timeline of smallpox immunization test with vaccinia virus vaccine-coated MN patches. (**b**) Photos showing the application of the vaccine-coated MN patch on the shaved dorsal side of experimental animals (BALB/C mouse) for smallpox vaccination. (**c**) Photos and SEM images of vaccine-coated MN patches before and after application, respectively (scale bar, 200 μm). (**d**) Titers obtained from vaccine-coated MN patches before and after inoculation (*n* = 7). The virus titers were measured through plaque assay after dissolving the coated layer by immersing the MN patch in PBS. (**e**) Neutralizing antibody titers at 3 and 6 weeks after vaccination. Serum samples of female BALB/C mice were collected at 3 and 6 weeks after vaccination (*n* = 7). (**f**) The vaccinia virus specific T-cell immune response was measured at 12 weeks after MN-mediated vaccination (*n* = 10). Statistical significance compared to the no treatment sample was determined by a *t*-test (* *p* < 0.05, ** *p* < 0.01).

**Figure 5 vaccines-10-00561-f005:**
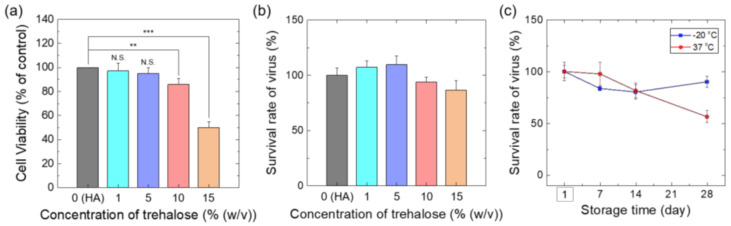
(**a**) Cytotoxic effect of trehalose concentrations in HA coating solutions on vero cells (*n* = 3). (**b**) Survival rate of vaccinia virus in HA coating solutions with different trehalose concentrations (*n* = 3). (**c**) Long-term stability test of vaccine-coated MN patches stored at different temperature (−20 °C and 37 °C). Relative virus titer (compared to as prepared vaccine-coated MN patch) examined through plaque assay after dissolving the coated layer by immersing the MN array in PBS (*n* = 6). Statistical significance compared to the 0% trehalose sample was determined by a *t*-test (** *p* < 0.01, *** *p* < 0.001).

## Data Availability

The data that support the findings of this study are available from the corresponding authors upon reasonable request.

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
