# Peer review of "Low-Temperature Multiple Micro-Dispensing on Microneedles for Accurate Transcutaneous Smallpox Vaccination"

_vaccines, 2022, doi:10.3390/vaccines10040561_

Round 1

Reviewer 1 Report

The authors use replicating poxviruses coated to microneedles for the immunization of mice.

Although the approach might be very useful for mass vaccination there are critical restrictions in understanding and use.

1: The second-generation smallpox vaccine (CJ-50300, HK innoN, Korea) must be described and referenced as to how it was derived.

2: Even though the in vitro analysis of coated viral particles is reassuring that not massive denaturation of virus occurs (Fig. 3), the delivery of viral particles into the skin is more difficult to evaluate. A scarification with known amounts of virus must be used as a control. This control should provide the same antibody titer and T cell response in mice compared to the new approach.

3: The epidermis and dermis of mice are extremely thin. Any depilatory cream is damaging and thus disturbing the anatomy of the skin. At the base of the tail towards the back, the hair of mice can be plugged manually and cream omitted. These controls must be made.

4: Antibody neutralization titers and T cell responses do not really indicate the degree of protection against a challenge with a deadly mouse poxvirus like ectromelia. This is a “must” and would provide indications together with control experiments (2, 3) as to how well this microneedle approach works.

5: The discussion must be broadened to a use of this technique to non replicating vaccines and possible other applications like prick tests to probe for allergies.

6: Considering the death rate and severe side effects of replicating vaccinia vaccines in healthy young soldiers in the USA, it seems unlikely that South Korea would use this vaccine in a population at large. This must be mentioned in the article and alternative pox viral vaccines at least mentioned; see ref.  (1), (2), (3).

  1. Casey CG, Iskander JK, Roper MH, Mast EE, Wen XJ, Torok TJ, Chapman LE, Swerdlow DL, Morgan J, Heffelfinger JD, Vitek C, Reef SE, Hasbrouck LM, Damon I, Neff L, Vellozzi C, McCauley M, Strikas RA, Mootrey G. 2005. Adverse events associated with smallpox vaccination in the United States, January-October 2003. JAMA 294:2734-43.
  2. Poland GA, Grabenstein JD, Neff JM. 2005. The US smallpox vaccination program: a review of a large modern era smallpox vaccination implementation program. Vaccine 23:2078-81.
  3. Halsell JS, Riddle JR, Atwood JE, Gardner P, Shope R, Poland GA, Gray GC, Ostroff S, Eckart RE, Hospenthal DR, Gibson RL, Grabenstein JD, Arness MK, Tornberg DN, Department of Defense Smallpox Vaccination Clinical Evaluation T. 2003. Myopericarditis following smallpox vaccination among vaccinia-naive US military personnel. JAMA 289:3283-9.

Reviewer 2 Report

Even almost finished vaccination target of "smallpox Vaccine", but microneedles transdermal vaccination is quite useful and interesting for smallpox vaccine, due to specific target depth beyond skin barrier.

Also, interesting information about coating method, viscosity conditions, HA concentrations as well as trehalose concentrations are very useful and technically important.

Q1. To coat vaccination uniformly on the tip of microneedles, surface energy of needle itself (more detail information about plasma treatment necessary) are important as well as shapes and dimension of needle itself.  Can explain in detail surface treatment conditions of needle?  Did you try these coating way with different shapes of needle (cones, pyramidal shapes, etc.)?

Q2. How long time it take to fabricate one patch (including total process time of dispensing, coating, drying, etc.)?

Q3. To increase enough dose of vaccination in total needle patch area comparing with dissolvable needle way, coated way should be improved somehow or the suggested way is enough?

Round 2

Reviewer 1 Report

Not required